# Effects of Different Processing Methods of Coffee *Arabica* on Colour, Acrylamide, Caffeine, Chlorogenic Acid, and Polyphenol Content

**DOI:** 10.3390/foods11203295

**Published:** 2022-10-21

**Authors:** Olga Cwiková, Tomas Komprda, Viera Šottníková, Zdeněk Svoboda, Jana Simonová, Jan Slováček, Miroslav Jůzl

**Affiliations:** 1Department of Food Technology, Faculty of AgriSciences, Mendel University in Brno, 613 00 Brno, Czech Republic; 2Research Institute of Brewing and Malting, Lípová 511/15, 120 00 Praha, Czech Republic

**Keywords:** colour assessment, GC/MS chromatography, degree of roasting, Maillard reaction

## Abstract

An effect of a processing method (dry and wet) and a degree of roasting (light, medium, and dark) of 15 coffee (*Coffea arabica*) samples on the content of caffeine, chlorogenic acid (CQA), total polyphenols (TPP), acrylamide (AA), and on the colour parameters L*, a*, and b* was evaluated. Neither processing nor roasting affected caffeine content (*p* > 0.05). The degree of roasting accounted for 46% and 72% of explained variability of the CQA content and AA content, respectively (*p* < 0.05). AA content was in the range from 250 (wet-processed, light-roasted samples) to 305 µg·kg^−1^ (wet-processed, dark-roasted coffees), but the dark roasting only tended (*p* > 0.05) to increase AA content. Wet-processed, dry-roasted coffee had higher (*p <* 0.05) TPP content (48.5 mg·g^−1^) than its dry-processed, dry-roasted counterpart (42.5 mg·g^−1^); the method of processing accounted for 70% of explained variability of TPP. Both the method of processing and the degree of roasting affected the L*, a*, and b* values (*p* < 0.01), but the lower values (*p* < 0.05) of these parameters in the dark-roasted samples were found only within the wet processing. A negative correlation between the AA content and lightness (L*) was established (r = −0.39, *p* < 0.05). It was concluded that from the consumers’ viewpoint, the results of the present study indicate relatively small differences in quality parameters of coffee irrespective of the method of processing or degree of roasting.

## 1. Introduction

Coffee is one of the most famous plantation products [1]. It is grown in more than 70 countries [2]. The best-known commercially grown species include *Coffea arabica*, *Coffea canephora,* and *Coffea liberica* [3,4]. The resulting quality of coffee is influenced not only by the coffee variety but also by the processing method [5]. Post-harvest processing has a great influence on the chemical composition of coffee beans [6]. Currently, three methods of coffee bean isolation are mainly used: dry (natural), semi-dry, and wet-processed [7]. In the dry method, cherries are dried up to a moisture content of 10–12% without prior treatment. The skin is mechanically removed only after drying. In the semi-dry method, cherries are deskinned and then dried in the sun without fermentation. In the wet method, the cherries are first deskinned, followed by natural fermentation. The beans are then washed and dried in the sun [8].

The roasting of coffee beans is considered one of the most important and critical phases of bean processing because during the roasting process, green coffee beans are transformed into roasted beans with a characteristic taste and aroma [9] and the formation of aromatic compounds, colour, and grain texture changes significantly [10]. This process could be divided into three main types, light-roast, medium-roast, and dark-roast. Then the final roasting process will affect the taste, colour, and smell of the coffee beans [1]. During the roasting, some substances contained in coffee beans, such as chlorogenic acid (CQA), may be altered or degraded [11]. CQA is a naturally occurring phenolic compound widely distributed in plants [12]. These changes, most often in chemical composition, depend on both the variety and the degree of roasting. According to Belguidoum et al. [13], the caffeine content of green coffee beans can be variable. It depends mainly on the type of coffee and its variety, geographical region, and processing. Caffeine is a thermostable substance that does not change even if it is over-roasted [11,14]. However, the high temperatures used in coffee roasting result in the loss of less caffeine by sublimation [15].

Several toxic substances, e.g., acrylamide (AA), furan, and 5-hydroxymethylfurfural, are produced by the Maillard reaction during coffee roasting [16]. The amount of AA in coffee depends on the type and the roasting conditions. Equally important is the sorting of immature beans as they contain significantly more free asparagine compared to mature beans [17]. AA has been classified by the International Agency for Research on Cancer (IARC) as a probable human carcinogen, mainly affecting the nervous system [18,19].

There exists not much literature dealing with the correlation between coffee ground colour and caffeine, CQA, and TPP content. Tsai and Jioe [20] found a high correlation between the L* parameter (lightness), resp. a* parameter (the colour space between green and red), and the CQA content of ground coffee. However, they found no relationship between caffeine content and colour space in ground coffee.

The objective of the present study was to evaluate the effect of the method of processing and the degree of roasting, respectively, on the caffeine, CQA, total polyphenols, AA content, and colour parameters of 15 different coffee samples and to assess possible relationships between the above-mentioned chemical and sensorial parameters.

## 2. Materials and Methods

### 2.1. Materials

Fifteen samples of coffee of the *Coffea arabica* variety originating from fourteen different countries (Brasil, Colombia, Costa Rica, Cuba, Dominican Republic, Ethiopia, Guatemala, Honduras, India, Kenya, Mexico, Nicaragua, Peru, and Uganda) were evaluated regarding chemical (caffeine, chlorogenic acid, total polyphenols, and acrylamide) and colour (CIELab coordinates) parameters.

Two following variability factors were considered in the study design: the processing method (wet and dry) and the degree of roasting (light, medium, and dark). The coffee cherries designated in the country of origin as dry-processed were instantly set to dry within sunshine until reaching a water content of 15%; after being separated, the coffee beans were dried again until they had a moisture content of 12%. Wet processing involved the mechanical depulping of coffee cherries, separating the ripe fruit, and following the fermentation step; the resultant parchment coffee was dried to produce the green coffee beans.

Samples of wet- and dry-processed coffee beans, respectively, were roasted in three coffee roaster companies located in three cities in the Czech Republic (Brno, Jihlava, and Telč, respectively). The roasting temperatures were basically the same in all three roasters: light roasting of 194–197 °C, medium roasting of 202–209 °C, and dark roasting of 212–217 °C. Due to the trade secret, the roasters declined to communicate more details regarding roasting conditions. Three light-, three medium-, and four dark-roasted samples within the set of the wet-processed samples were produced; two light-, two medium-, and two dark-roasted samples were prepared as far as the dry processing is concerned. Samples of the roasted coffee beans were obtained from the roasters packaged in plastic bags weighing approximately 1 kg each.

### 2.2. Caffeine and Chlorogenic Acid (CQA) Determination

The following standards and chemicals were used: chlorogenic acid (Dr Ehrenstorfer GmbH, Augsburg, Germany), caffeine (Sigma-Aldrich, Prague, Czech Republic), methanol for HPLC (Sigma-Aldrich, Prague, Czech Republic), and acetic acid 100% for HPLC (Sigma-Aldrich, Prague, Czech Republic).

The sample of ground coffee (7 g) was added into a beaker (250 mL), and the caffeine and CQA were coextracted for 5 min with 100 mL of hot (90 °C) demineralised water. Then the sample was filtered through the paper filter. Coffee extracts were analysed by the HPLC method described in Yoe-Ray et al. [21] with the following modifications. Caffeine and CQA were separated on the column Kinetex, Biphenyl, 150 × 4.6 mm, particle size 5 µm (Phenomenex, Torrance, CA, USA) with the guard column (SecurityGuard ULTRA Cartridge, UHPLC Biphenyl, for 4.6 mm ID columns, Phenomenex) using a liquid chromatograph Agilent 1260 Infinity II (Agilent Technologies, Santa Clara, CA, USA) with quaternary pump and diode array detector. The column was thermostated at 35 °C. Methanol/acetic acid 1% (20:80 *v*/*v*) was used as a mobile phase. The flow rate of the mobile phase was 1 mL min^−1^. The separated CQA and caffeine, respectively, were detected at 273 nm. The calibration curve was used for the quantitative determination; the calibration curve points ranged from 10 to 100 µg mL^−1^.

### 2.3. Total Polyphenols Determination

The following chemicals were used: Folin–Ciocalteu reagent (Penta, Czech Republic), sodium carbonate p.a. (Lach-Ner, Neratovice, Czech Republic), and ferulic acid (Sigma-Aldrich, Czech Republic).

The sample of ground coffee (7 g) was added into a beaker (250 mL), and polyphenols were extracted for 5 min with 100 mL of hot (90 °C) demineralized water. Then the sample was filtered through the paper filter. The total polyphenols in the coffee extracts were determined spectrophotometrically by the Folin–Ciocalteu method according to a procedure described by Dae-Ok et al. [22] with the following modifications. Twenty µL of the coffee extract was transferred to a 2 mL vial and diluted with 980 µL of demineralized water. In total, 200 µL of the diluted coffee extract was mixed with 1 mL of Folin–Ciocalteu reagent (diluted ten times: 10 mL of Folin–Ciocalteu reagent and 90 mL of demineralized water). The solution was left to react for 10 min, then 800 µL of 7.5% Na_2_CO_3_ (*w*/*v*) was added, and the solution was mixed. After 30 min, the absorbance of the solution was measured against a blank sample at 765 nm. The ferulic acid calibration curve was used to quantitatively determine the total polyphenols. The calibration curve points ranged from 0.02 to 0.1 mg·mL^−1^.

### 2.4. Determination of Acrylamide (AA)

The following standards and chemicals were used: AA (1 mg·mL^−1^) in methanol (Absolute Standards, Hamden, CT, USA); (^13^C_3_)acrylamide (1 mg·mL^−1^) in methanol (Cambridge Isotope Lab, Inc., Tewksbury, MA, USA); 2,3-dibromopropionamide (1 mg·mL^−1^) in methanol (Absolute Standards); bromine, potassium bromide, hydrobromic acid, saturated bromine water, sodium thiosulphate, and triethylamine (all from Merck, Darmstadt, Germany); methanol and ethylacetate (purity for high-performance liquid chromatography; Sigma-Aldrich, Prague, Czech Republic).

Acrylamide in the coffee samples was quantified by the GC–MS method. Before the GC–MS analysis, acrylamide was derivatized by bromation in order to prepare a more volatile and less polar compound with better mass characteristics that can be easily extracted from aqueous solutions and better detected by the GC–MS method.

Ten mL of internal standard [(^13^C_3_)acrylamide] and 50 mL of distilled water (60 °C) were added to the sample of ground coffee (10 g). After sonification (20 min) in an ultrasonic bath, the homogenized sample was quantitatively transferred into a centrifuge tube and centrifuged at 5000 rpm for 30 min. Five ml of the supernatant was pipetted into a glass flask, and potassium bromide (2 g) and hydrobromic acid (resulting pH 0–1) were added. After cooling the mixture in the refrigerator (30 min), 2 mL of bromine water was added. Content in the flask was mixed, and the flask was placed in a container with crushed ice for 10 h in the refrigerator. During bromination, acrylamide was converted to 2,3-dibromopropionamide. After bromination, the excess bromine was titrated with sodium thiosulphate (1 mol·mL^−1^ solution) till discolouration. The flask content was transferred into a centrifugal tube. Ethyl acetate (5 mL) was added to the tube, and the suspension was shaken for 3 min and afterwards centrifuged at 2000 rpm for 5 min. After centrifugation, 1 mL of the organic phase was transferred with a pipette to a plastic microtube, and triethylamine (0.2 mL) was added, resulting in unstable 2,3–dibromopropionamide converting to a more stable derivate 2-bromopropenamid. Microtubes were then shaken for 15 min and then centrifuged at 2000 rpm for 5 min. Subsequently, the tube content was transferred to a 2 mL glass vial.

The gas chromatograph Trace GC Ultra Finnigen with the mass detector Trace DSQ Thermo Finnigen (Arcade, NY, USA) was used to determine acrylamide bromine derivate content. The separation was carried out on the DB-WAX capillary column (30 m × 0.25 mm × 0.25 µm; J&W Scientific, Folsom, CA, USA), with helium as a carrier gas with a flow rate of 1.5 mL·min^−1^. The column temperature was held at 50 °C for 1 min and then increased at 15 °C·min^−1^ to 150 °C for 5 min. The PTV (Programmable Temperature Vaporizing) injector temperature was 200 °C, and the injector worked in the splitless mode (1 min). The transfer line temperature was 200 °C. The mass selective detector operated in selected ion monitoring (SIM) mode with positive electron impact (EI) ionization.

2-bromopropenamide was identified based on the retention time and specific ions *m/z* 149 and *m/z* 151. Quantification was performed using a calibration curve. To achieve reliable results and maximum selectivity, isotopically marked acrylamide (1,2,3 -^13^C_3_ acrylamide) was used as an internal standard. Specific ions of 2-bromo(^13^C_3_)propenamide were *m/z* 152 and *m/z* 154.

The calibration curve was linear in the range from 75 to 1550 μg·kg^−1^ (r^2^ = 0.9998). The linear scope was sufficient for the analysis of the real samples.

### 2.5. Colour Parameters Measurement

The CIE colour parameters of ground coffee were measured according to Dong et al. [18] using a Minolta Chroma meter (CM-3500d; Minolta Company, Osaka, Japan). Triplicate analyses were made for each sample. Results were expressed as L*, a*, and b* values (CIELAB coordinates). The L* represents lightness (100 for white and 0 for black), and chromatic coordinates a* and b* represent the colour space between red (+a*) and green (−a*) and between yellow (+b*) and blue (−b*), respectively. The total colour difference (∆E*_ab_) was calculated according to Kulapichitr et al. [23].

### 2.6. Statistical Analysis

Differences in the chemical and colour parameters, respectively, between samples of coffee processed by two methods (dry and wet) and consequently roasted to three different degrees (light, medium, and dark) were evaluated by one-factor ANOVA with post hoc Tukey’s test. Moreover, a combination of a given processing method and a given degree of roasting (e.g., dry-processed, medium-roasted sample) was considered one factor. The differences between means at *p* < 0.05 were considered significant.

Consequently, a processing method and a degree of roasting were considered to be two independent variables (factors); their effect on the chemical and colour parameters of the coffee samples was assessed by two-factor ANOVA for an interaction. Relationships between all tested parameters were evaluated using Pearson’s correlation. STATISTICA 14 software (StatSoft, Prague, Czech Republic) was applied for all evaluations.

## 3. Results

### 3.1. Caffeine, Chlorogenic Acid, Polyphenols, and Acrylamide Content

The content of caffeine, chlorogenic acid, total polyphenols, and acrylamide in the tested coffee samples processed by either wet or dry method and consequently roasted to three different degrees is shown in Table 1. Effects of the two tested variability factors (processing method and degree of roasting) on the content of the above-mentioned chemical parameters are presented in Table 2.

As shown in Table 1, neither the processing method nor the degree of roasting had a significant effect on caffeine content in the coffee samples (*p* > 0.05). Wet-processed coffee roasted to the highest degree (dark) had higher CQA content (*p* < 0.05) in comparison with the dry-processed, medium-roasted samples. When calculated irrespective of the production method, medium-roasting decreased (*p* < 0.05) CQA content from 10.4 to 6.9 mg·g^−1^ in comparison with light-roasting (data not shown in Table 1).

There was higher total polyphenol content (*p* < 0.05) found in wet-processed, dark-roasted coffee samples in comparison not only with dry-processed, dark-roasted but also with the dry-processed, medium-roasted counterparts. When calculated irrespective of the degree of roasting (data not shown in Table 1), wet processing increased total polyphenol content (*p* < 0.05) in comparison with dry processing (from 42.7 to 46.8 mg·g^−1^), and dark roasting increased (*p* < 0.05) total polyphenol content in comparison with medium roasting, when the method of processing was not considered.

As far as acrylamide content is concerned, light-, medium-, and dark-roasted coffee samples processed by the wet method did not differ (*p* > 0.05) from their dry-processed counterparts (Table 1). However, nevertheless, dark roasting increased (*p* < 0.05) acrylamide content (303.3 µg·kg^−1^) as compared to both medium (260.6 µg·kg^−1^) and light roasting (260.1 µg·kg^−1^) when the production method was not considered (data not shown in Table 1).

As it is apparent from Table 2, neither the processing method nor the degree of roasting significantly affected caffeine content in coffee (*p* > 0.05); the percentage of the explained variability accounted for by these two factors was negligible despite significant interaction processing × roasting (*p* = 0.01). The effect of roasting was important in the case of chlorogenic acid (*p* = 0.03), as this factor accounted for nearly half of the explained variability in the content of this phenolic compound. The effect of roasting had an even higher effect on the production of acrylamide: it participated in nearly 72% of the explained variability of acrylamide content in coffee (*p* = 0.02). On the other hand, the processing method was much more important regarding total polyphenols, accounting for more than two-thirds of explained content variability (*p* = 0.001).

### 3.2. Colour Characteristics

Differences in the colour parameters between samples of coffee processed by two methods and consequently roasted to three different degrees are shown in Table 3. The calculated effects of the two above-mentioned independent variables (method of processing and degree of roasting) on the CIELab coordinates are presented in Table 4.

The dry-processed, dark-roasted coffee samples were lighter (higher L*-value; *p* < 0.05; Table 3) than the wet-processed, dark-roasted counterparts. The same was true regarding both a* and b* coordinates (higher values in the dry-processed, dark-roasted samples in comparison with the wet-processed, dark-roasted counterparts; *p* < 0.05). Higher values of both L* and a* and b* (*p* < 0.05) were found in the light-roasted coffees compared to the medium- or dark-roasted ones when the data were evaluated irrespective of the processing method (data not shown in Table 3).

As shown in Table 4, both the processing method and the degree of roasting significantly (*p* < 0.05) affected all tested colour coordinates, and both independent variables accounted for a substantial percentage of explained variability. Moreover, significant (*p* < 0.05) interactions between the method of processing and the degree of roasting were found in all tested colour parameters.

### 3.3. Relationships between Chemical and Colour Parameters

The relationships between the tested parameters were calculated using correlation analysis, and the results are presented in Table 5.

It follows from Table 5 that apart from the tight, logical relationships between the three colour coordinates, only two more significant correlations were found: acrylamide content was in a negative relationship with the level of lightness (L*; r = 0.39; *p* < 0.05), and content of total polyphenols was strongly correlated with the content of chlorogenic acid (r = 0.46; *p* < 0.05). On the other hand, caffeine content has no significant relationship with any other tested parameter; it only tended to correlate with acrylamide content (r = 0.32; *p* = 0.07).

## 4. Discussion

### 4.1. Caffeine

No significant effect of the processing method on caffeine content was found in the present study. Regarding the method of processing, it is necessary to mention that the coffee samples used in our study were delivered to the roasters from the particular countries of origin only with a designation either “dry” or “wet” processed, without any other details describing post-harvest processing. Therefore, we cannot reliably confirm that our data, on the one hand, are contrary to the findings of Król et al. [24]. They reported that temperature and storage time affected the caffeine content in roasted coffee beans, with caffeine content increasing with longer storage time. On the other hand, we cannot reliably confirm that our data agree with the results of Worku et al. [25] or Jeon et al. [26] that caffeine content was not related to post-harvest processing. The latter conclusion is corroborated by the results of Diego et al. [27] and Duarte, Pereira, and Farah [6]; they reported no significant effect of wet or dry processing on the caffeine content.

Similar to the processing method, no significant effect of a degree of roasting on the caffeine content was established in the present study. This is rather surprising because caffeine solubility in water increases with temperature, and a drag by water vapour released during roasting causes the loss of caffeine [28]. According to Franca et al. [28], an approximate reduction of 30% in caffeine content occurs during roasting. Different roasting conditions affected the caffeine content in the study by Hečimović et al. [29]. Furthermore, Król et al. [24] found that the quantity of caffeine was significantly affected by the roasting degree: light-roasted coffee had higher caffeine levels (6.42 mg g^−1^) than medium- (5.77 mg g^−1^) and dark-roasted (2.63 mg g^−1^) beans. Irrespective of the above-mentioned differences, caffeine content reported by Król et al. [24] was substantially lower in comparison with our data (12–13 mg g^−1^, Table 1). A slightly higher caffeine content in dark-roasted as compared to light-roasted coffee was reported Tsai and Jioe [20].

On the other hand, our data regarding an insignificant effect of roasting degree on the caffeine content are in agreement with the study by Jeon et al. [30] or Alamri et al. [31]. They also did not notice any effect of the degree of roasting on caffeine content in coffee beans, and a possible reason is the thermal stability of caffeine [32].

### 4.2. Phenolic Substances: Chlorogenic Acid, Total Polyphenols

Despite the fact that the only significant difference in the CQA content between the coffee samples analysed in the present study regarded higher CQA content in the wet-processed and dark-roasted coffee in comparison with the dry-processed and medium-roasted samples (Table 1), an effect of roasting on the CQA content was significant. It accounted for 46% of the explained variability of this trait (Table 2). The roasting process of coffee beans leads to the degradation of CQA isomers to volatile components and causes their transformation to quinic acid, its epimer, and four CGA lactones [16]. As a consequence, roasting decreases the antioxidant activity of coffee beans, which depends on the structural integrity of phenolic compounds, including CQA [33,34]. However, antioxidant activity has not been measured in the present study.

The process of roasting resulted in a decrease in CQA concentration in coffee beans in a study by Farah et al. [35]. Moreover, Mills et al. [36] found out that the greater the degree to which the coffee has been roasted, the lower the content of CQA. Similarly, Tsai and Jioe [20] reported a decrease in CQA after roasting and the lowest content was reached in dark-roasted coffee samples. We can only partially confirm these findings [27,30]: when calculated irrespective of the production method, medium roasting significantly decreased CQA content in comparison with light roasting in the present study, but a manifestation of the dark roasting was less conspicuous.

As far as a method of processing is concerned, no difference in CQA content between dry- and wet-processed coffee samples was established in the present study, which disagrees with the results of Balyaya and Clifford [37], who reported lower levels of total CQA in the wet-processed *Arabica* coffee in comparison with dry-processed samples.

For a comparison of the absolute values, an average CQA content in *Coffea arabica* reported by Bresciani et al. [38] was 5.89 mg g^−1^, which corresponds to the dry-processed, medium-roasted coffee samples established in our experiment (Table 1).

However, as far as total polyphenol content is concerned, the average data presented by Bresciani et al. [38], approximately 16 mg g^−1^, are substantially lower in comparison with our results (42–49 mg g^−1^, Table 1). Because TPP was determined by the same method (Folin–Ciocalteau assay) both in the quoted experiment [38] and in the present study, the differences are likely a consequence of a different matrix: silverskin (i.e., inner fruit layer surrounding coffee beans; [38]) and whole (roasted) coffee beans (present experiment).

Though CQA is an important component of phenolic substances in coffee beans, an effect of a processing method and a degree of roasting on CQA and TPP, respectively, was very different in the present experiment. Contrary to CQA, the effect of processing on total polyphenols was highly significant, accounting for more than two-thirds of the explained variability in TPP (Table 2). Wet processing increased total polyphenol content (*p* < 0.05) in comparison with dry processing when the data were calculated irrespective of the degree of roasting.

On the other hand, an effect of a degree of roasting on TPP content was insignificant in the present study, which disagrees with both the data of Hečimović et al. [29] and Król et al. [24], who identically reported the highest content of total polyphenolic compounds in coffees roasted in light and medium conditions. Similarly, Bobková et al. [39] also found the highest levels of polyphenols in light-roasted coffees. However, when the discrepancy between the results of the quoted experiments [24,29,35] and the present study was analysed in more detail, it follows from Table 1 that the TPP content in the dry-processed, light-roasted samples tended to be higher in comparison with both dry-processed, medium-roasted and dry-processed, dark-roasted counterparts. Furthermore, the TPP content in the wet-processed, light-roasted coffees tended to be higher than in wet-processed, medium-roasted samples. Therefore, the whole discrepancy was caused only by higher (significantly or in a tendency) TPP content in wet-processed, dark-roasted coffees in comparison with all remaining samples; however, the latter data are difficult to explain.

### 4.3. Acrylamide

Fifteen samples of coffee originating from fourteen countries were analysed in the present experiment. However, because we have no control over the production conditions in particular countries, including the concentration of asparagine in coffee beans, it would be groundless to evaluate AA content in the coffee samples based on the country of origin. So only a range of AA content in the analysed coffee samples is relevant and can be compared with similar studies. The range of the AA values was 230–360 µg·kg^−1^ in the present experiment, which fully corresponds with the results obtained by Svoboda et al. [40], where AA values in ground coffee moved in the range from 240 μg·kg^−1^ to 358 μg·kg^−1^. The Czech Agriculture and Food Inspection Authority reported that from 2005–2011, AA content in roasted ground coffees ranged from 87 to 460 μg·kg^−1^ [40], a wider range of AA content in comparison with the present study. Similarly, Andrzejewski et al. [41] found AA concentrations in the ground coffee ranged from 40 to 400 μg·kg^−1^. According to the EU Commission database, roasted coffee finished products have a median AA level of 265–290 µg·kg^−1^ [42]. AA levels in roasted *Arabica* coffee never exceeded the limit of 400 μg·kg^−1^ in a study by Vezzulli et al. [43].

Generally speaking, the post-harvest coffee processing type determines the coffee quality [6], and according to Giulia et al. [44], all processing stages can play an important role in the final AA levels in the product. However, regarding AA content in the present experiment, the processing method accounted for less than one per cent of the explained variability (Table 2). The average AA content in coffee processed by the dry and wet method in the present study was very similar: 276 and 272 μg·kg^−1^, respectively.

Contrary to the method of processing, the degree of roasting had a decisive effect on the AA content in the present experiment, accounting for 72% of the explained variability (Table 2). Temperature and time are important process parameters affecting the Maillard reaction and, thus, possible AA formation in foods [45]. Guenther et al. [42] found that light-roasted coffees may contain higher amounts of AA than very dark-roasted beans because only 20% of the initially formed AA survives the roasting process, as reported by Lantz et al. [46]. Anese et al. [47] also confirmed higher content of AA in medium-roasted in comparison to dark-roasted coffee. The same conclusion was reached by Granby and Fagt [48]. In addition, Schouten et al. [49] found the highest AA levels (730 µg·kg^−1^) in the *Arabica* samples at the light- and medium-roasting degree. However, the results of the present experiment are contrary to these findings [42,46,47,48,49]: when all data regarding the AA content were analysed irrespective of the method of processing, higher (*p* < 0.05) AA content (303 µg·kg^−1^) in the dark-roasted coffees in comparison with medium-roasted (261 µg·kg^−1^) or light-roasted (260 µg·kg^−1^) samples was established. Similar results were also reported by Wawrzyniak and Jasiewicz [50]. A possible explanation for the higher AA content in the dark-roasted coffees may be the longer storage periods of the light- and medium-roasted coffee samples, leading to a decrease in AA levels in comparison to the shorter storage times of the dark-roasted coffees before the samples were analysed. However, this hypothesis was not possible to confirm in the present experiment. Nevertheless, according to Delatour et al. [51], AA is not stable in commercial coffee stored in its original container, and losses of 40 to 60% have been recorded in coffees stored at room temperature [46]. Moreover, Lachenmeier et al. [52] reported that the presence of defective beans could also lead to a relatively higher AA content in roasted coffee because unripe beans contain significantly higher concentrations of free asparagine than ripe ones, which may also account for the higher AA content in dark-roasted coffee.

Moreover, using longer roasting and storage times seems to be an inappropriate way to decrease AA content in coffee due to the possible formation of other toxic compounds (e.g., 5-hydroxymethylfurfural, furan, and furfuryl alcohol), which may have a negative effect on human health and reduction in desired organoleptic properties of the final beverage [49].

### 4.4. Colour Coordinates

It is rather surprising that the method of processing accounted for more explained variability in lightness (L*; 43%) than the degree of roasting (34%) in the present experiment (Table 4). It is likely a consequence of the higher L*-value of the dry-processed, dark-roasted coffee samples in comparison with the wet-processed, dark-roasted counterparts (Table 3). Nevertheless, the roasting degree was also established as an important lightness influencing factor in the present study, though only in the wet-processed coffee samples. The L*-value of the wet-processed, light-roasted coffees was higher than both the medium-roasted, wet-processed and dark-roasted, wet-processed samples. It partially corresponds with the conclusions of Bicho et al. [53] and Schouten et al. [49] that lightness (L*) decreases with an increasing roasting intensity as a consequence of the development of higher brown colour intensity. Processing methods produce various colour-influencing substances, including melanoidins [54] and polymeric macromolecular compounds formed by the Maillard reaction, giving processed foods such as coffee a dark brown colour [55].

According to Tsai and Jioe [20], the dry fermentation method turns the raw coffee beans from white to brown, while the wet fermentation method turns them from blue to green. Based on the results of the present study (Table 3), the dry-processed, dark-roasted coffees, compared to wet-processed, dark-roasted samples, had higher both the a*-value and the b*-value (stronger transition from red to green and from yellow to blue, respectively).

### 4.5. Relationships between the Tested Variables

Because the strong relationships (*p* < 0.05) between the tested colour parameters logically follow their definitions within the CIELAB colour space [56], one of the more interesting correlations found in the present study is the negative relationship between acrylamide content and the instrumentally measured lightness (L*; r = −0.39; Table 5). Moreover, acrylamide formation by the Maillard-type reaction depends on the overall thermal input (superposition of temperature and baking time [57]). Thus, higher thermal input explains, on the one hand, a higher acrylamide content and, on the other hand, a darker surface of the product (i.e., lower L*-value). We observed a similar relationship between acrylamide content in gingerbread and the L*-value (r = −0.49) in our previous experiment [58].

However, relationships between the AA content in coffee and its colour described in the available literature are ambiguous. In accordance with our data, Mojska and Gielecińska [59] reported a negative correlation between AA levels in roasted coffee and intensity of colour, but contrary to our findings, Schouten et al. [49] found a positive correlation between AA level and the L* parameter in ground roasted coffee. Moreover, Akgün et al. [60] reported a weak positive correlation between the AA contents and the L*-values of Turkish coffees.

As follows from Table 5, chlorogenic acid content (r = 0.46) accounted for a substantial part of the observed variability in total polyphenol content in the present study. Generally speaking, correlation cannot be interchanged with a causal relationship. However, in this case, the statement that CQA content explained a considerable part of total polyphenols is substantiated: CQA, an ester of caffeic acid and quinic acid, refers to a general polyphenol family of esters [61]. Though the coefficient of correlation between CQA and total polyphenol content was significant in the present study, the relationship was nevertheless much weaker in comparison with the results of the study by [62], who reported correlation coefficients between total polyphenols and content of 3-CQA, 4-CQA, and 5-CQA in sweet potatoes at the level of 0.93, 0.72, and 0.81, respectively.

No significant relationship between CQA content and colour characteristics was found in the present study, contrary to the results of Tsai and Jioe [20], who reported a strong correlation between CQA × L* and CQA × a*, respectively.

## 5. Conclusions

The present study evaluated fifteen samples of coffees originating in fourteen different countries, processed by two different methods (wet or dry processing) and roasted to three different degrees (light, medium, and dark). The parameters of interest were chemical substances typical for coffee (caffeine and phenolic substances), including toxicologically important acrylamide and instrumentally measured colour coordinates. Due to the fact that the detailed production conditions of coffee were not possible to control, the effect of only two independent variables on the above-mentioned parameters was considered: the method of processing and the degree of roasting. Neither of these factors affected the caffeine content in the coffee samples. The two tested phenolic substances reacted differently on processing/roasting: total polyphenols were significantly affected only by the method of processing, and chlorogenic acid only by the degree of roasting. However, no clear trend of increasing the degree of roasting from light to medium to dark was established either in the CQA content or in the TPP content.

Similarly, in acrylamide, though the degree of roasting accounted for more than two-thirds of the explained variability in its content, no unequivocal trend of an increasing degree of roasting on the AA content was found in the present study. Only dark roasting tended to increase AA content within both tested methods of processing. However, the AA content in all tested coffee samples was below the level of 400 µg·kg^−1^.

Both the method of processing and the degree of roasting significantly affected the tested colour characteristics, but decreasing the L*, a*, and b* values in the direction from light to medium to dark was apparent only in the wet-processed samples.

So, from the viewpoint of coffee consumers, the results of the present study indicate relatively small differences in quality parameters typical for coffee due to the method of processing and the degree of roasting. It regards the positive traits (phenolic substances with antioxidant capacity), the negative markers (acrylamide content), and the colour characteristics.

## Figures and Tables

**Table 1 foods-11-03295-t001:** Caffeine, chlorogenic acid (CQA), total polyphenols (TPP), and acrylamide (AA) content in the coffee samples were processed by either wet or dry method and consequently roasted to three different degrees: light, medium, and dark, respectively.

		Parameter (Mean ± SEM)			
		Caffeine (mg·g^−1^)	CQA (mg·g^−1^)	TPP (mg·g^−1^)	AA (µg·kg^−1^)
Processing Method	Roasting Degree				
Dry (*n* = 10)	light (*n* = 4)	12.0 ± 0.16 ^a^	9.0 ± 1.59 ^ab^	43.1 ± 2.01 ^ab^	275.2 ± 12.23 ^a^
	medium (*n* = 3)	11.9 ± 0.25 ^a^	5.4 ± 0.67 ^b^	42.5 ± 0.93 ^b^	255.7 ± 16.03 ^a^
	dark (*n* = 3)	13.0 ± 0.08 ^a^	8.5 ± 0.12 ^ab^	42.5 ± 0.63 ^b^	297.2 ± 10.31 ^a^
Wet (*n* = 6)	light (*n* = 2)	12.4 ± 0.38 ^a^	8.1 ± 1.15 ^ab^	47.8 ± 1.50 ^ab^	250.0 ± 9.42 ^a^
	medium (*n* = 2)	12.9 ± 0.17 ^a^	7.9 ± 0.95 ^ab^	43.4 ± 0.96 ^b^	263.9 ± 10.81 ^a^
	dark (*n* = 2)	12.0 ± 0.40 ^a^	10.8 ± 0.94 ^a^	48.5 ± 1.24 ^a^	305.0 ± 18.19 ^a^

Notes: a,b: means with different superscripts in columns differ at *p* < 0.05; one-way ANOVA with post hoc Tuckey’s test. TPP are expressed as ferulic acid equivalent.

**Table 2 foods-11-03295-t002:** Effect of the processing method and the degree of roasting, respectively, on caffeine, chlorogenic acid (CQA), total polyphenols (TPP), and acrylamide (AA) content in the coffee samples processed by either wet or dry method and consequently roasted to the three different degrees; two-factor ANOVA with interaction; *n* = 16.

Parameter	Effects of the Variability Factors								
	Processing			Roasting			Processing × Roasting		
	F-Value	% of the Explained Variability	*p*-Value	F-Value	% Explained Variability	*p*-Value	F-Value	% of the Explained Variability	*p*-Value
Caffeine	0.12	2.0	0.73	0.29	4.8	0.75	4.73	77.0	0.01
CQA	2.22	25.3	0.14	4.03	46.0	0.03	1.51	17.0	0.23
TPP	12.01	70.2	0.001	2.24	13.1	0.12	1.87	10.9	0.17
AA	0.06	0.9	0.81	4.61	71.8	0.02	0.75	11.7	0.48

**Table 3 foods-11-03295-t003:** Values of the CIELAB coordinates L*, a*, b* in the coffee samples processed by either wet or dry method and, consequently, roasted to three different degrees, light, medium, and dark, respectively. L* represents lightness (100 for white and 0 for black), chromatic coordinates a* and b* represent the colour space between red (+a*) and green (−a*), and between yellow (+b*) and blue (−b*), respectively.

Colour Coordinates (Mean ± SEM)				
Processing Method	Roasting Degree	L*	a*	b*
Dry	light (*n* = 4)	28.1 ± 0.15 ^a^	11.2 ± 0.33 ^abc^	14.3 ± 0.64 ^ab^
	medium (*n* = 3)	25.9 ± 0.67 ^ab^	10.1 ± 0.41 ^bcd^	11.7 ± 0.83 ^bc^
	dark (*n* = 3)	28.4 ± 0.04 ^a^	11.8 ± 0.02 ^a^	15.6 ± 0.04 ^a^
Wet	light (*n* = 2)	28.0 ± 0.80 ^a^	11.6 ± 0.10 ^ab^	14.9 ± 0.30 ^a^
	medium (*n* = 2)	24.6 ± 0.38 ^b^	9.2 ± 0.23 ^d^	9.7 ± 0.46 ^c^
	dark (*n* = 2)	24.5 ± 0.78 ^b^	9.9 ± 0.47 ^cd^	11.8 ± 0.94 ^bc^

Notes: a–d: means with different superscripts in columns differ at *p* < 0.05; one-way ANOVA with post hoc Tuckey’s test.

**Table 4 foods-11-03295-t004:** Effect of the processing method and the degree of roasting, respectively, on L*, a*, and b* coordinates in the coffee samples processed by either wet or dry method and, consequently, roasted to the three different degrees. L* represents lightness (100 for white and 0 for black), chromatic coordinates a* and b* represent the colour space between red (+a*) and green (−a*), and between yellow (+b*) and blue (−b*), respectively; two-factor ANOVA with interaction; *n* = 16.

Colour Coordinate	Effects of the Variability Factors								
	Processing			Roasting			Processing × Roasting		
	F-Value	% of the Explained Variability	*p*-Value	F-Value	% of the Explained Variability	*p*-Value	F-Value	% of the Explained Variability	*p*-Value
L*	10.14	43.0	0.002	8.02	34.3	0.001	4.25	18.2	0.02
a*	7.99	30.2	0.007	12.40	46.1	0.001	5.09	19.5	0.001
b*	8.50	29.1	0.005	15.00	51.3	0.001	4.71	16.1	0.01

**Table 5 foods-11-03295-t005:** Relationships between tested chemical markers: caffeine, chlorogenic acid (CQA), total polyphenols (TPP) and acrylamide (AA), and colour coordinates: L* (lightness), a* (colour space between red and green) and b* (colour space between yellow and blue); correlation coefficients, *n* = 16.

Tested Variable	CQA	TPP	AA	L*	a*	b*
Caffeine	−0.06	0.20	0.32	−0.08	−0.27	−0.23
CQA		0.46 *	0.27	0.04	0.25	0.29
TPP			0.30	0.08	0.01	0.08
AA				−0.39 *	−0.30	−0.24
L*					0.82 *	0.85 *
a*						0.99 *

* Significant correlation (*p* < 0.05).

## Data Availability

Data is contained within the article.

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
