# Peer review of "Effects of Different Processing Methods of Coffee Arabica on Colour, Acrylamide, Caffeine, Chlorogenic Acid, and Polyphenol Content"

_foods, 2022, doi:10.3390/foods11203295_

Round 1
Reviewer 1 Report
Overall, in my opinion the design of the presented study does not allow to fulfil its aim (“to evaluate the effect of processing method on the caffeine, CGA, total polyphenols and AA content of coffee from different production areas”). The study only allows to evaluate and compare the quality of 15 coffee samples included in this study, without concluding on the effect of (a) country of origin, (b) processing, (c) roasting. Thus, with the current title, abstract, methodology and results description, and finally conclusions.
Another general remark: plenty of small language/grammatic/style/editorial corrections are necessary throughout the manuscript.
Title: Should be rewritten to address the above mentioned problem. And, in addition, since chlorogenic acid belongs to polyphenols, I would suggest to think about revising the title to avoid impression that the authors do not consider this phenolic acid as one of the polyphenols.
Methodology: references for all the analytical methods used in the study should be provided. Any modifications should be mentioned. “The standards and chemicals used” should be listed not only for acrylamide determination, but also for all other analyzes (ideally together in one section). Statistical methods applied (one-way ANOVA) also indicate that the samples were simply compared among each other (without taking into account processing factors such as dry/wet processing and roasting).
Results and discussion:
Line 169: It is incorrect to say that caffeine is one of the main constituents of the coffee beans.
Line 171-173: Coffee from different countries is being compared here, while not necessarily the country of origin, but maybe the differences in coffee processing are responsible for the differences described here (which cannot be assessed in this study, due to its design). And opposite – when the effect of processing is described further in the manuscript, it’s in fact not necessarily the effect of processing. This is the problem of the whole Results and Discussion section.
List of references:
Should be formatted following the Foods journal style (the information provided, e.g. journal names should be abbreviated; and the font style, spaces between rows etc. – there is a dedicated style for references in the Foods manuscript template and this one should be used).
Author Response
Dear editor, dear reviewers,
thank you for the note, the design of the experiment, including the name of the submitted article, was modified.
The article was redesigned by the author's team, where we invited the Professor Tomas Komprda, who expanded our team. He was included among co-authors.
We appreciate the work you have devoted and thank you very much for your reviews based on your review. It allowed us to improve the manuscript and evaluate the work of all authors.

Reviewer 2 Report
Please see comments in attached file

Author Response

(The authors gave the same response as above.)

Reviewer 3 Report
Title: The title is not clear and should be rephrased. Possible rephrased titles may be: “Effect of different processing methods on Colour, Acrylamide, Caffeine, Chlorogenic Acid and Polyphenol Content of Arabica Coffee”. Also, the study, in my opinion, explains more about the differences based on geographical origin rather than processing, so the authors should consider changing the title accordingly.
Keywords: Most of the keywords resemble the title and should be substituted with more particular ones to increase the search index compatibility.
Line 54: The authors mention “several toxic substances” but only give one example. State some more examples?
Introduction: This section should be improved by including previous literature, gap in knowledge, novelty and clear objectives of this study.
Section 2.1: To better understand the effects of processing conditions, some information on coffee processing should be provided.
Table 1: It is unclear in terms of the footnote “A, B different coffee roasting degree” then why a separate column “type of roast” is included and what about the roast degree of other coffee samples because this footnote is mentioned only on samples 7 and 14??
Furthermore, the other footnote “1, 2, 3 Different coffee roasting manufactory” does not have much significance, rather, information on processing conditions would add value to understanding the processing effects?
Line 184-186: Not clear whether the authors are describing the study by “Fujioka and Shibamoto [26]” or their own study??
Section 3.1.1: The authors have explained the results of other studies more than their own. The results need to be explained critically and comprehensively.
Also, as the authors have stated that several studies have established no to minimal effect on the caffeine content post-processing or roasting, then what was the need to carry out a kind of repetitive research? The authors need to explain the need for this analysis.
Table 1: In chlorogenic acid column the superscript letters are the same for both samples 3 and 5 even though the values are quite different. Check ??
Line 244: The authors claim that there is no difference in the CAG content as a result of processing methods, but if you see the CAG content of samples individually, there is clearly a significant difference in the processing methods. In my opinion, taking means and comparing is not the correct way, as this takes into consideration other factors like geographical location, variety and roasting level and not just the processing method. Rather the use of 2 way-anova or other statistical tools would provide better conclusions.
Also, results need to be explained in a more critically way with information on the changes taking place or the probable reasons for the changes??
Line 262: The authors state that “no difference was found in total polyphenol content between light and dark–roasted ground coffee samples” but there is a difference between medium and dark roasting levels, as clear from Table 3, which the authors missed to point out and explain the reasons for that. Take into account all the changes and explain accordingly.
Polyphenolic content was evaluated using ferulic acid as a standard, so the results should be expressed as ferulic acid equivalent? Please check and correct throughout the manuscript
Line 272-274: Provide reasons for the effect of processing on the polyphenols?
Section 3.2: The first paragraph should begin with the findings of your study, and the first paragraph written in the manuscript should be added somewhere after line 299 to better explain the results and make it more clear for the readers.
Figure 1: If possible, include the pictures of other samples too, just 2 samples do not provide anything valuable. I guess the authors have added a figure just for the sake of adding a figure!!
Line 323: Adding a correlation table would provide more information than just stating there is no correlation and stating two values.
Line 326-330: Not required in the results and discussion section
Line 413-416: As pointed out earlier, the study details are more based on the geographical origin than processing, so the authors need to reconsider the objective and title or reconsider the approach to explaining the results and discussion
Line 416-419: This information does not relate to the objectives of this study. It seems to be random information added
Conclusion: Again, the authors begin this section with a focus on the geographical origin of the samples. The authors need to reconsider the approach or need to be more clear with their objectives and what they want to investigate.
I strongly agree with the authors that it is important to establish a relationship between the quality parameters and the processing of coffee. But, as several studies have failed to achieve this objective, so a different approach or experimental design is required by planning research taking only one or two factors and not many factors as done in this study. Taking a standard and uniform coffee sample from one location and then processing it under controlled parameters with varying magnitudes would provide a better picture of the effect of processing on the quality parameters.
I would also suggest authors make a significant revision in the level of English throughout the manuscript.
References: This study contains 70 references, which is excessive in my opinion for a research article. This seems to be an article discussing the results of other similar studies. Only selective references should be stated.
Author Response

(The authors gave the same response as above.)

Round 2
Reviewer 1 Report
Unfortunately vast majority of my concerns still remain valid and have not been addressed by the Authors. Maybe because addressing them would require total rewriting of the text and applying completely different analytical approach (or changing perspective). I will repeat that in the current "shape" the design of the presented study does not allow to fulfil its current aim. And with the data that the Authors possess, there is no way to appropriately answer the question about the effect of roasting or processing on the measured parameters. The study only allows to evaluate and compare the quality of 15 coffee samples included in this study, without concluding on the effect of processing or/and roasting.
Reviewer 2 Report
Hi dear
I Think all of my suggestion almost consider.
Reviewer 3 Report
The authors have revised the manuscript thoroughly.